# A Retrospective Study of Antimicrobial Resistant Bacteria Associated with Feline and Canine Urinary Tract Infection in Hong Kong SAR, China—A Case Study on Implication of First-Line Antibiotics Use

**DOI:** 10.3390/antibiotics11091140

**Published:** 2022-08-23

**Authors:** Olivia S. K. Chan, Myriam Baranger-Ete, Wendy W. T. Lam, Peng Wu, Michelle Yeung, Elaine Lee, Helen Bond, Owen Swan, Hein Min Tun

**Affiliations:** 1The School of Public Health, Li Ka Shing Faculty of Medicine, The University of Hong Kong, 7 Sassoon Road, Pokfulam, Hong Kong SAR, China; 2Asia Veterinary Diagnostics Laboratories, 22/F West Gate Tower, 7 Wing Hong Street, Cheung Sha Wan, Kowloon, Hong Kong SAR, China; 3Agriculture, Fisheries and Conservation Department, 5/F, Cheung Sha Wan Government Offices, 303 Cheung Sha Wan Road, Kowloon, Hong Kong SAR, China; 4The Hong Kong Veterinary Association, Hong Kong SAR, China

**Keywords:** canine and feline patients, antimicrobial resistance, multi-drug resistance

## Abstract

Urinary tract infection (UTI) is a common clinical diagnosis for which empirical antibiotics are used in veterinary medicine. For veterinarians, the description of canine and feline antibiograms can help with making prudent use decisions and guideline formulation. For public health officers and epidemiologists, a urinary antibiogram overview helps track and trend antimicrobial resistance (AMR). There is currently a knowledge gap in AMR prevalence associated with urinary tract infection in feline and canine patients and the resistance percentage of these microbes against some of the over-the-counter antibiotics available to local pet owners. This study has two aims. First, it aims to investigate the frequency of the bacteria and bacterial-resistance pattern in urine samples obtained from feline and canine patients. Second, it aims to determine the resistance of *Escherichia coli* (*E. coli*), the most frequently isolated bacteria, to first-line antibiotics. Results: We identified the five most-frequently isolated bacterial species and determined these isolates’ antibiotic sensitivity and resistance. The most-frequently isolated bacteria in feline and canine patients was *Escherichia coli* (*E. coli*). *E. coli* was identified, on average, in 37.2% of canine and 46.5% of feline urine samples. Among feline urinary samples, *Enterococcus* (14.7%) and *Staphylococcus* (14.5%) spp. were isolated more frequently, followed by *Pseudomonas* (4.8%) and *Klebsiella* (5.2%) spp. (). In canine samples, *Proteus* (17.9%) and *Staphylococcus* (13.2%) spp. were isolated more frequently, followed by *Enterococcus* (10.0%) and *Klebsiella* (8.59%) spp. Among these isolates, 40 to 70% of *Staphylococcus* spp. bacterial isolates from feline and canine patients were resistant to amoxicillin and ampicillin. During the three-year study period, among canine patients, 10 to 20% of *Staphylococcus* spp. bacterial isolates were resistance to fluoroquinolones, other quinolones, and third-generation cephalosporins. Among feline patients, 10% of *Staphylococcus* spp., 15 to 20% of *E. coli*, 50 to 60% of *Klebsiella* spp., and 90% of *Pseudomonas* spp. were resistant to cefovecin, a commonly used antibiotic.

## 1. Introduction

The term antimicrobial refers to any pharmaceutical, chemical, biological, or physical agent that stops or slows bacterial, viral, fungal, or microparasitic growth. Antimicrobials that target bacteria are commonly termed antibiotics. The dilemma in the use of antimicrobials, or antibiotics to treat bacterial infection, is the simultaneous ability of bacteria to evolve and develop resistance [1]. The cumulative or acute escalation of such resistance reduces the effectiveness of antibiotics as a treatment option. Therefore, there are guidelines and good practice recommendations around prudent antimicrobial use established by international entities, such as the International Society for Companion Animal Infectious Disease (ISCAID) and World Small Animal Veterinary Association (WSAVA) [2,3,4,5]. All guidelines indicate that antibiotics treatment should ideally be based on in vitro susceptibility testing, clinical history, and symptoms [6]. However, empirical antibiotics treatment is common, particularly when urinary signs of dysuria, anuria, pollakiuria, or stranguria are presented [7]. Therefore, information on the prevalence of antibiotic resistance can help veterinarians make informed decisions on empirical use and also bring about awareness of possible single- and multi-drug resistance. Such knowledge helps reduce misuse, improper use, or overuse of antimicrobials [8,9].

*Escherichia coli* (*E. coli*) is a commonly occurring uropathogen in cats and/or dogs, according to studies in China [10,11], the United States of America [12], Canada [13], Germany [14], Sweden [15], and Australia [16]. Less frequently occurring bacteria in urine samples include *Enterococcus faecalis*, *Staphylococcus*, and *Proteus* spp. [8,12,17,18,19]. The patterns of isolates and their resistance prevalence appear to vary geographically and temporally [20,21].

Rising antibiotic resistance trends in *E. coli* were observed in cats and dogs in Canada, China, and Poland. In Poland, there was a statistically significant increase in multi-drug-resistant *E. coli* between 2007 and 2013 [22]. In Canada, *E. coli* increased resistance to amikacin, amoxicillin-clavulanate, and cephalexin between 1994 and 2003 [23]. In China, *E. coli* demonstrated increased multi-drug resistance between 2012 and 2017 [24]. Concerned with the clinical observation of an upward antibiotic resistance trend locally, this study was conducted to investigate frequently isolated bacteria and the bacterial resistance pattern in urine samples obtained from feline and canine patients and to identify *E. coli* resistance to first-line antibiotics.

## 2. Results

A total of 15,449 urine samples from feline and canine patients were processed between 2018 and 2020. In 2018, 5787 samples were processed, of which 2196 samples grew bacteria, and 3591 did not grow bacteria. In 2019, 4670 samples were processed, and 1752 samples grew bacteria, and 2918 did not. In 2020, 4992 samples were processed, of which 1783 grew bacteria, and 3209 did not. In total, 5731 samples were positive for bacterial isolation, while 9718 of these samples did not yield bacterial growth.

Of all isolates, 3719 isolates were from canine and 2012 isolates were from feline patients. A total of 399 (11.0%) isolates were susceptible to all antibiotics tested, with 198 isolates from canine and 201 isolates from feline patients.

### 2.1. Antimicrobial Resistance in Commonly Identified Isolates in Feline and Canine Patients

Between 2018 and 2020, the most-frequently isolated bacteria in feline and canine patients was *Escherichia coli* (*E. coli*). *E.*
*coli* was identified, on average, in 37.2% of canine and 46.5% of feline urine samples. The frequency of the remaining bacterial species differed between cats and dogs. Among feline urinary samples, the following species were identified, in descending frequency: *Enterococcus* (14.7%), *Staphylococcus* (14.5%), *Pseudomonas* (4.8%), and *Klebsiella* (5.2%) spp. (Table 1). In canine samples, *Proteus* (17.9%) and *Staphylococcus* (13.2%) spp. were isolated the most frequently, followed by *Enterococcus* (10.0%) and *Klebsiella* (8.59%) spp. (Table 2).

Feline urine isolates demonstrated a distinct antibiotic resistance pattern in *E. coli*, *Enterococcus*, *Klebsiella*, *Pseudomonas*, and *Staphylococcus* spp. (Table 3). Among the *E. coli* samples tested, 40% were resistant to amoxycillin and ampicillin, and 15 to 20% were resistant to cephalexin, cefovecin, cefpodoxime, ceftriaxone, cephalothin, ciprofloxacin, enrofloxacin, enrofloxacin, marbofloxacin, ofloxacin, and trimethoprim-sulfamethoxazole. Among the *Enterococcus* isolates, about 90% were resistant to gentamicin and trimethoprim-sulfamethoxazole. About 35% of these *Enterococcus* isolates were resistant to ciprofloxacin, enrofloxacin, marbofloxacin, ofloxacin, doxycycline, and meropenem. Among the *Klebsiella* isolates, about 90% were resistant to amoxycillin and ampicillin. About 50% to 60% of these *Klebsiella* isolates were resistant to cephalexin, cefovecin, cefpodoxime, ceftriaxone, cephalothin, chloramphenicol, ciprofloxacin, enrofloxacin, marbofloxacin, ofloxacin, doxycycline, nitrofurantoin, and trimethoprim-sulfamethoxazole. Among the *Pseudomonas* isolates, about 90% were resistant to amoxycillin, amoxycillin-clavulanate, ampicillin, cephalexin, cefovecin, cefpodoxime, ceftriaxone, cephalothin, doxycycline, nitrofurantoin, and trimethoprim-sulfamethoxazole. About 25% of these *Pseudomonas* isolates were resistant to ciprofloxacin, enrofloxacin, marbofloxacin, and ofloxacin. Among the *Staphylococcus* isolates, about 45% were resistant to amoxycillin, ampicillin, and penicillin. About 10% of these *Staphylococcus* isolates were resistant to clindamycin, cephalexin, cefovecin, cefpodoxime, ceftriaxone, cephalothin, chloramphenicol, ciprofloxacin, enrofloxacin, marbofloxacin, ofloxacin, doxycycline, meropenem, and trimethoprim-sulfamethoxazole.

Canine urine isolates demonstrated a different antibiotic resistant pattern in *E. coli*, *Enterococcus*, *Klebsiella*, and *Staphylococcus* spp. (Table 3). For the *E. coli* isolates, 70% of the isolates tested against clindamycin were resistant, and 40% tested against amoxycillin and ampicillin were resistant. About 15% of the *E. coli* isolates were resistant to a number of cephalosporins, including cephalexin, cefovecin, cefpodoxime, ceftriaxone, and cephalothin, and the same percentage was resistant to fluoroquinolone (such as ciprofloxacin, enrofloxacin, marbofloxacin, and ofloxacin), doxycycline, and nitrofurantoin. For the *Enterococcus* isolates, about 90% canine samples demonstrated resistance to gentamicin and nitrofurantoin. About 40% of these *Enterococcus* isolates were resistant to doxycycline, and 30% were resistant to fluoroquinolones, ciprofloxacin, enrofloxacin, marbofloxacin, and ofloxacin. About 20% of these *Enterococcus* isolates were resistant to amoxycillin, amoxycillin-clavulanate, ampicillin, and meropenem. Among the *Klebsiella* isolates, 90% were resistant to amoxycillin and ampicillin; about 60% were resistant to clindamycin; and about 30% were resistant to cephalexin, cefovecin, cefpodoxime, ceftriaxone, cephalothin, chloramphenicol, ciprofloxacin, enrofloxacin, marbofloxacin, ofloxacin, doxycycline, nitrofurantoin, and trimethoprim-sulfamethoxazole. About 20% of these *Klebsiella* isolates were resistant to ticarcillin-clavulanate and piperacillin. For the *Proteus* isolates, about 90% of the isolates were resistant to doxycycline and nitrofurantoin, and 20% were resistant to chloramphenicol, amoxycillin, and ampicillin. Among the *Staphylococcus* isolates, about 70% were resistant to amoxycillin, ampicillin, and penicillin. About 60% were resistant to doxycycline; 30% to trimethoprim-sulfamethoxazole; and 20% to meropenem, ofloxacin, marbofloxacin, enrofloxacin, ciprofloxacin, chloramphenicol, cephalexin, cefovecin, cefpodoxime, ceftriaxone, and cephalothin.

Of the total of 5731 samples positive for bacterial isolation, 3216 isolates (56%) exhibited resistance to more than three antibiotics classes (multi-drug resistance). Of these multi-drug resistant isolates, 2187 (68.0%) isolates were from canine and 1029 (32.0%) isolates were from feline patients (Table 4).

### 2.2. Resistance of E. coli from Urine of Feline and Canine Patients to First-Line Antibiotics

The sensitivity and resistance profile of *E. coli* isolates to first-line and non-first-line antibiotics are listed in Table 5.

ISCAID, AVMA, AVA, and BVA provide antibiotic guidelines for veterinarians in Hong Kong, China. The ISCAID guideline recommends amoxicillin and trimethoprim-sulmethoxazole as first-line antibiotics for initial or empirical treatment [2]. About 40% of urinary *E. coli* isolates were resistant to amoxicillin and 15% to trimethoprim-sulmethoxazole in feline and canine patients. Furthermore, 40% of urinary *E. coli* isolates were resistant to the non-first-line antibiotic doxycycline and 15% to ofloxacin, marbofloxacin, enrofloxacin, ciprofloxacin, chloramphenicol, cephalothin, ceftriazone, cefopodoxime, and cefovecin. Approximately 20% of canine and 15% of feline *E. coli* isolates were resistant to a number of non-first-line antibiotics, including doxycycline, ofloxacin, marbofloxacin, enrofloxacin, ciprofloxacin, cephalothin, ceftriazone, cefopodoxime, and cefovecin (Table 5).

### 2.3. E. coli Antibiotic Resistance Trend in Feline and Canine Species

The first- and non-first-line antibiotic antimicrobial resistance pattern has a significant implication regarding clinical empirical use [19]. Between 2018 and 2020, feline *E. coli* resistance against the first-line antibiotic amoxicillin increased from 44.4 to 49.0%, while trimethoprim-sulfamethoxazole resistance remained the same. In the same period, feline E. coli resistance against the non-first-line antibiotic quinolones ciprofloxacin, enrofloxacin, marbofloxacin, and doxycycline increased by 3%, 5%, and 2%, respectively (Figure 1). In 2018, the canine baseline *E. coli* resistance against first-line antibiotics was higher than in feline patients. During this period, canine *E. coli* resistance against first-line amoxicillin increased by 3%, while trimethoprim-sulfamethoxazole reduced by 5%. In the same period, canine *E. coli* resistance against the non-first-line antibiotic quinolones ciprofloxacin, enrofloxacin, marbofloxacin, and doxycycline reduced by 5%, 2%, 4%, and 7%, respectively (Figure 2).

## 3. Conclusions

The pattern of the most-frequently isolated bacteria and antibiotic resistance in feline and canine patients were similar except in two bacteria. The difference was observed in *Klebsiella* spp., which was isolated more frequently in feline patients, and *Pseudomonas* spp., which was identified more frequently in canine patients. For both groups of patients, *E. coli*, the most frequently isolated bacteria, carried a similar resistance pattern to first-line antibiotics such as amoxycillin and trimethoprim-sulfamethoxazole. Additionally, the results showed that *E. coli* is two-times-more resistant to amoxycillin compared to trimethoprim-sulfamethoxazole. The results also showed that *E. coli* is resistant to some commonly used but non-first-line antibiotics, such as fluoroquinolone and doxycycline. 

## 4. Discussion

This study highlighted AMR as a threat to small animal patients in the Hong Kong Special Administrative Region of China, helps inform public health stakeholders of AMR’s zoonotic and anthropozoonotic potential, and establishes the need for responsible pet antimicrobial consumption among pet carers and owners. This study provided two clinical implications. First, the monitoring of antibiotic resistance updates clinicians on potential antibiotic resistance in their patients. This information helps facilitate clinicians’ prudent antimicrobial prescription, especially in considering the weight of empirical antibiotic use. Second, resistance monitoring, especially among first-line antimicrobials, helps the profession establish and evaluate their clinical antimicrobial guidelines. Additionally, it is important that antibiotic treatment continues to be conducted according to antibiograms in small animal veterinary practice. In parts of Asia, and particularly in small animal medicine, such AMR information deserves iteration and update.

Studies in different localities can provide a basis for comparison, encourage AMR stewardship collaboration, and strengthen international veterinary AMR dialogue. This is particularly true, as antibiotic sensitivity and resistance can be similar or varied depending on the patient’s species, location of lesion, and physical settings [25]. For instance, a study in Spain demonstrated similar findings to this study, in that canine patients had a higher rate of positive urinary cultures than felines [26]. The same study also indicated similar findings of *E. coli*, *Proteus*, *Staphylococcus*, and *Enterococcus* spp. frequently isolated in canine patients, while *E. coli*, *Enterococcus*, and *Staphylococcus* spp. were frequently identified in feline patients [27]. There are numerous contemporary publications and projects that emphasize the importance of collaboration and sustainable efforts to mitigate AMR [28,29]. Continued inter-sectoral and international cooperation and sustainable partnership are needed to implement prudent antimicrobial use and AMR stewardship in small animal veterinary medicine [30,31].

In the public health and One Health frameworks, AMR monitoring in small animals helps identify potential zoonotic and anthropozoonotic agents [32,33,34,35,36,37] For example, the consistent investigation of clinically relevant Gram-positive and Gram-negative bacteria, including the “ESKAPE” bacteria, helps clinicians and public health officers establish appropriate antibiotic use guidelines [38,39,40,41,42,43]. Thus, continuous monitoring of antibiotic resistance among locally common bacteria has important clinical relevance and public health implications.

## 5. Materials and Methods

### 5.1. Informed Consent

Data were collected and stored as advised by the research ethics committee and according to the privacy guidelines set out by the University of Hong Kong. This study did not involve the use of animals.

### 5.2. Data Collection

Data were collected from a commercial laboratory between 2018 and 2020. The specimens had been collected from specialist clinics and general practice clinics in Hong Kong. The laboratory was accredited and followed the Clinical and Laboratory Standards Institute (CLSI) protocol. Data inclusion criteria were (1) canine or feline urinary sample antibiograms; (2) samples had to be from the locality of Hong Kong; and (3) urine samples had to be analyzed in the laboratory between January 2018 and December 2020.

Multi-drug resistance was defined as bacteria that showed resistance to at least three different antibiotic classes [44]. The multi-drug resistance antimicrobial class categorization made reference to the World Association for Animal Health (OIE) list of antimicrobial agents of veterinary importance, WSAVA’s Essential Medicine List, and World Health Organization (WHO) ACCESS pharmaceutical framework.

A total of 10% of the data were entered a second time to check for accuracy and validity. Data description was conducted by R software.

### 5.3. Data Management

The information obtained from the laboratory’s in-house database was manually transferred to an Excel spreadsheet. Variables were entered by month and year of isolation, host species, bacterial species, and profiles of antimicrobial susceptibility. Case identifiers were delinked from the data.

### 5.4. Collection of Urine Samples

Urine specimens were inoculated within three hours upon receipt.

### 5.5. Isolation and Identification of Bacteria

Urine samples were streaked using an inoculation loop onto CHROMID^®^ CPS^®^ Elite (CPSE) (BioMerieux) plates. Each plate was streaked with 1 uL of the sample horizontally and vertically, covering all of the plate medium.

The sample was first incubated for 16 to 24 h. The first review was conducted at 18 to 24 h. *E. coli* was identified by positive chromogenic results, as indicated by the growth and color representation (http://www.biomerieux-culturemedia.com/product/9-chromid-cps-elite, access on 20 August 2022). If isolates other than *E. coli* were identified, a matrix-assisted laser desorption/ionization-time of flight (MALDI-TOF) mass spectrometry analysis was conducted for bacteria identification.

A subculture was conducted when mixed chromogenic results and first plating failed to provide a pure isolated colony for the identification and antimicrobial test. In this case, the clearest colony was identified from the mixed growth on the CPSE plate and sub-cultured to a new CPSE plate to be re-incubated for identification.

Colony forming unit (CFU) counting was conducted in all urine specimens. The colony counts were established at the 16th to 24th h. Ranges were categorized as 1 to 9 colonies (10^3^ to 10^4^ CFU/mL), 10 to 99 colonies (10^4^ to 10^5^ CFU/mL), and more than or equal to 100 colonies (>10^5^ CFU/mL). After the first review, all samples were subjected to second incubation for another 18 to 24 h.

### 5.6. Culture and Antimicrobial Sensitivity Test

After identification, an antimicrobial susceptibility test was conducted by the Kirby–Bauer disk diffusion method and by interpreting zones of growth inhibition according to CLSI.

A standard panel of antimicrobials was tested against all positive cultures. The standard panel included amikacin, amoxycillin-clavulanate, ampicillin, cefovecin, cefpodoxime, ceftriaxone, cephalexin, cephalothin, chloramphenicol, ciprofloxacin, clindamycin, doxycycline, enrofloxacin, gentamicin, marbofloxacin, nitrofurantoin, ofloxacin, piperacillin-tazobactam, and trimethoprim-sulfamethoxazole. Extended panel antimicrobials included imipenem and carbepenem. An extended antimicrobial panel was also performed for all *staphylococcus* spp. for methicillin susceptibility. Second panels were conducted for methicillin-resistant *staphylococcus* spp., including fusidic acid, mupirocin, rifampicin, and vancomycin.

## Figures and Tables

**Figure 1 antibiotics-11-01140-f001:**
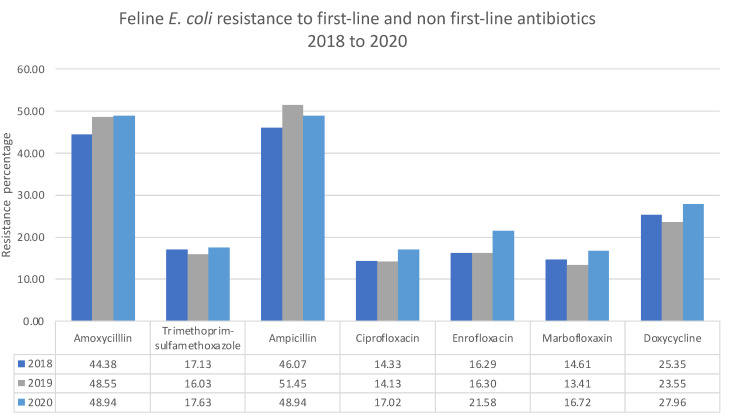
Resistance of *E. coli* from urine of feline patients to first-line and non-first-line antibiotics.

**Figure 2 antibiotics-11-01140-f002:**
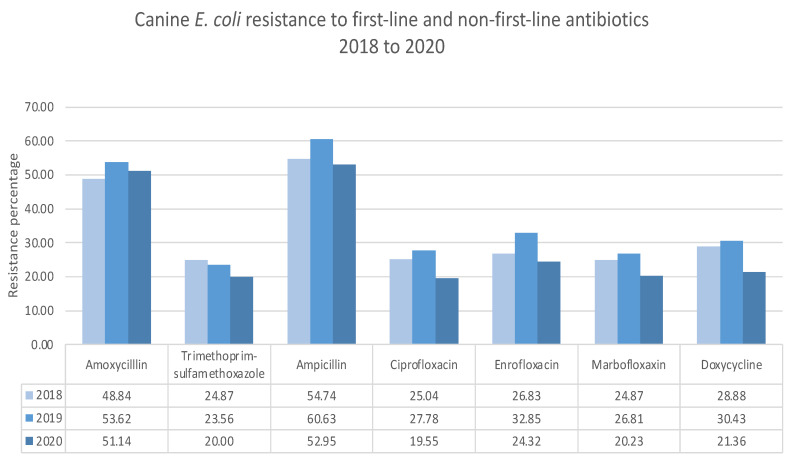
Resistance of *E. coli* from the urine of canine patients to first line and non-first-line antibiotics.

**Table 1 antibiotics-11-01140-t001:** Bacteria isolated from feline urine samples in Hong Kong between 2018 and 2020.

Canine Patients	Number and Percentage (%) of Isolates by Year			
Isolates	2018	2019	2020	Total Isolates (By Bacteria)
*Escherichia coli **	559 (37.34)	414 (34.85)	440 (39.32)	1413
*Proteus* spp. ***	249 (16.63)	234 (19.7)	193 (17.25)	676
*Staphylococcus* spp.	222 (14.83)	142 (11.95)	144 (12.87)	508
*Enterococcus* spp.	142 (9.49)	136 (11.45)	103 (9.2)	381
*Klebsiella* spp. ***	122 (8.15)	102 (8.59)	101 (9.03)	325
*Streptococcus* spp.	76 (5.08)	68 (5.72)	60 (5.36)	204
*Pseudomonas* spp. ***	50 (3.34)	33 (2.78)	40 (3.57)	123
*Enterobacter* spp. ***	17 (1.14)	17 (1.43)	9 (0.8)	43
*Citrobacter Koseri **	10 (0.67)	Not isolated	8 (0.71)	18
*Corynebacterium* spp.	7 (0.47)	10 (0.84)	Not isolated	17
Total isolates (By year)	1454	1156	1098	3708

* Gram negative microbe.

**Table 2 antibiotics-11-01140-t002:** Bacteria isolated from canine urine samples in Hong Kong between 2018 and 2020.

Feline Patients	Number and Percentage (%) of Isolates by Year			
Isolates	2018	2019	2020	Total Isolates (By Bacteria)
*Escherichia coli **	356 (48.17)	276 (45.17)	329 (46.67)	961
*Enterococcus* spp.	93 (12.58)	103 (16.86)	103 (14.61)	299
*Staphylococcus* spp.	104 (14.07)	87 (14.24)	106 (15.04)	297
*Pseudomonas* spp. ***	31 (4.19)	30 (4.91)	38 (5.39)	99
*Proteus* spp. ***	29 (3.92)	28 (4.58)	36 (5.11)	93
*Klebsiella* spp. ***	43 (5.82)	26 (4.26)	38 (5.39)	107
*Streptococcus* spp.	27 (3.65)	20 (3.27)	10 (1.42)	57
*Enterobacter* spp. ***	12 (1.62)	13 (2.13)	21 (2.98)	46
*Pasturella* spp. ***	9 (1.22)	9 (1.47)	4 (0.57)	22
*Stenotrophomonas* spp. ***	2 (0.27)	3 (0.49)	2 (0.28)	7
*Corynebacterium* spp.	3 (0.41)	2 (0.33)	Not isolated	5
*Bacillaceae* spp.	2 (0.27)	1 (0.16)	1 (0.14)	4
*Morgenella* spp. ***	1 (0.14)	1 (0.16)	2 (0.28)	4
*Serratia* spp. ***	2 (0.27)	1 (0.16)	1 (0.14)	4
*Acinetobacter junii **	1 (0.14)	1 (0.16)	1 (0.14)	3
*Actinomyces* spp.	Not isolated	1 (0.16)	1 (0.14)	2
*Aerococcus viridans*	Not isolated	1 (0.16)	Not isolated	1
Total isolates (By year)	715	603	693	2011

* Gram negative microbe.

**Table 3 antibiotics-11-01140-t003:** Antimicrobial-resistance profile of major bacteria from urine of feline and canine patients.

		Bacterial Resistance to Antibiotics (%) in Feline Patients					Bacterial Resistance to Antibiotics (%) in Canine Patients				
*Antibiotic categories*	Antibiotics	*E. coli*	*Enterococcus* spp.	*Klebsiella* spp.	*Pseudomonas* spp.	*Staphylococcus* spp.	*E. coli*	*Enterococcus* spp.	*Klebsiella* spp.	*Proteus* spp.	*Staphylococcus* spp.
Aminoglycosides	Gentamicin	12	98	42	22	16	16	100	24	8	38
	Amikacin										
Carbapenems	Imipenem			2	9	16					
	Meropenem		38	2	8	16		30			20
Cephalosporins	Cephalexin	18	98	62	99	17	22	97	36	11	19
	Cefovecin	16	97	59	96	17	20	98	32	9	19
	Cefopodoxime	17	98	61	98	18	21	99	32	9	19
	Ceftriaxone	14	96	50	90	17	18	96	28	7	19
	Cephalothin	19	96	62	96	17	24	97	37	10	19
Fluoroquinolones	Ciprofloxacin	14	42	59	29	15	21	30	34	10	24
	Enrofloxacin	14	42	60	34	18	22	32	34	15	22
	Marbofloxacin	14	44	58	32	17	21	32	34	6	23
	Ofloxacin	14	44	58	34	19	21	34	34	7	22
Penicillins/Beta-lactamase inhibitors	Ticarcillin-clavulanic acid	6	40	36	19	18	8	36	22	1	19
	Piperacillin-tazobactam			12			2		4	1	
Penicillin	Amoxicillin	47	28	98	99	60	50	24	96	22	78
	Amoxicillin-clavulanate	7	28	38	99	16	10	24	26	8	28
	Ampicillin	48	28	98	99	60	52	24	96	21	78
	Penicillin		22	4		58		19			70
	Methicillin										
	Oxacillin					17					19
	Floxacillin				12	4					2
	Piperacillin	4		38	8		4		20	4	
	Chloramphenicol	10	24	50	99	14	12	26	30	30	26
	Doxycycline	23	38	50	88	17	24	42	37	98	60
	Nitrofurantoin	1	18	58	99		3	10	44	99	1
	Trimethoprim-sulfamethoxazine	14	97	52	90	17	22	95	38	28	36
	Rifampicin					17					
	Vancomycin							2			
	Mupirocin					2					
	Clindamycin	71	65	73	73	18	70	64	68	64	30
	Fusidic					2					5

**Table 4 antibiotics-11-01140-t004:** Multi-drug resistance in feline and canine urine samples.

	Resistant Isolates by Number (%) and Species		
No of Antimicrobial Class	Total No. of Isolates (%)	Feline No. of Isolates (%)	Canine No. of Isolates (%)
0	477 (8.14%)	243 (11.78%)	234 (6.15%)
1	1324 (22.59%)	524 (25.5%)	800 (21.03%)
2	843 (14.39%)	260 (12.65%)	583 (15.33%)
3	912 (15.56%)	237 (11.53%)	675 (17.74%)
4	598 (10.2%)	183 (8.91%)	415 (10.91%)
5	498 (8.5%)	145 (7.06%)	353 (9.28%)
6	387 (6.6%)	141 (6.86%)	246 (6.47%)
7	367 (6.26%)	134 (6.52%)	233 (6.13%)
8	307 (5.24%)	118 (5.74%)	189 (4.97%)
9	131 (2.24%)	65 (3.16%)	66 (1.74%)
10	16 (0.27%)	6 (0.29%)	10 (0.26%)

**Table 5 antibiotics-11-01140-t005:** First-line and non-first-line antibiotics resistance among *Escherichia coli* in feline and canine urine samples.

	Feline		Canine	
	Recommended Antibiotics by Classes	Resistance Percentage	Recommended Antibiotics by Classes	Resistance Percentage
First-line antibiotics	Amoxicillin for lower UTI	40%	Amoxicillin for lower UTI	40%
	Trimethoprim-sulfate for lower UTI	15%	Trimethoprim-sulfate for lower UTI	20%
	Fluoroquinolone for pyelonephritis *	15%	Fluoroquinolone for pyelonephritis *	20%
Non first-line antibiotics	Ceftriazone **	15%	Ceftriazone **	20%
	Cefopodoxime **	15%	Cefopodoxime **	20%
	Cefovecin **	15%	Cefovecin **	20%
	Ofloxacin *	15%	Ofloxacin *	20%
	Marbofloxacin *	15%	Marbofloxacin *	20%
	Enrofloxacin *	15%	Enrofloxacin *	20%
	Ciprofloxacin *	15%	Ciprofloxacin *	20%
	Chloramphenicol	15%	Chloramphenicol	10%
	Doxycycline	40%	Doxycycline	20%

* Quinolones are a highest priority critically important antibiotics for human medicine by 2018 6th Edition for risk management due to non-human use (World Health Organization). ** Third-generation cephalosporin are a highest priority critically important antibiotics for human medicine by 2018 6th Edition for risk management due to non-human use (World Health Organization).

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
