# Peer review of "A Retrospective Study of Antimicrobial Resistant Bacteria Associated with Feline and Canine Urinary Tract Infection in Hong Kong SAR, China—A Case Study on Implication of First-Line Antibiotics Use"

_antibiotics, 2022, doi:10.3390/antibiotics11091140_

Round 1

Reviewer 1 Report

The manuscript presents a wealth of data on the antimicrobial susceptibility of dog and cat urine isolates in Hong Kong. The topic is very important, which is why these data are valuable for publication. However, there are things for which I suggest corrections, please see the corrections in the attachment.
Kind regards,
Zrinka Štritof

Reviewer 2 Report

This study which reports on a retrospective analysis of antimicrobial resistance profile in bacteria associated with UTI in dogs and cats  in Hong Kong, China between 2018 and 2020 is of great interest to veterinary clinicians and public health. Though the manuscript is fairly well written, it still requires extensive editing to improve on the English.

The following specific comments should be noted:

A) Abstract:

1) AMR should be in relation to the bacteria/isolates from feline and canine patients and not as stated in relation to the patients. Thus, line 29 can be rephrased to read " AMR pattern in bacteria from feline.....Again, 64 % of feline and 66% of canine isolates.......

2) Reference to list established by WHO, OIE and SWAVA not necessary in the abstract.

3) Rephrase lines 32-35 in line with comments in 1 and 2 above for clarity.

4) Most frequently isolated bacteria should be stated in the abstract (especially as this is one of the objectives of the study).

B) Introduction

1) Lines 42-44 are not relevant.

2) Lines 45-47 should be at end of introduction as they capture the aim and importance of the study.

3) Lines 48-52 can be used to draw conclusion of the study (the study lacks conclusion).

4) Lines 64/65: "signs" preferable to "symptoms"

5) Lines 70-72 to be rephrased for clarity.

6) Line 79: MDR is among isolates from feline and canine patients.

C) Results

1) It is necessary to indicate the number of samples processed during the period under review (also per year understudy). The number of samples positive for bacterial isolation should also be stated.

2) Lines 81-88 need to be rephrased for clarity.   

3) What is shown as Figures 1 and 2 are actually tables. In these tables the total number (N) of samples processed or samples positive for bacterial growth and the number (n) for the respective year should be indicated. The title of Table 1 (your figure 1), line 90, can be rephrased thus: Bacteria isolated from feline urine samples in Hong Kong between 2018 and 2020. Same for Table 2. Please check the bacterium Acinebacter in Table 1

4) What is bug-drug resistance?

5) The legends in 'Figures 3 and 4' are very difficult to read. I suggest the figures be converted into a tables for better understanding.  Suggested titles of the tables: Antimicrobial resistance profile of major bacteria from urine of ..........

6) Line 143: Resistance of E. coli from urine of feline and canine patients to first line antibiotics.

7) 'Figure 5' (Table 5) presents only resistance profile and no sensitivity profile. This 'figure' shows Annual trends in AMR among E. coli isolates  from feline and canine species.

8)  Rephrase title of Figs 6 and 7 (suggested to be Figs. 1 and 2).

9) Rephrase line 175 for clarity.

10) Figure 8 should be Table.

11) Rephrase lines 181-182 for clarity.

12) I find it very difficult to understand the section covered by lines 183-197. Please rewrite for clarity.

13) What were the prevalent MDR pattern(s)? The MDR patterns should contain the specific antibiotics to which the isolates were resistance rather than the class of antibiotics. Stating the class suggests that the organisms were resistant to all the antibiotics within the specified class, which is quite misleading. MDR is resistance to at least one agent in three or more classes.

 D) Discussion

1) Information on lines 211 to 219 not really necessary.

2) lines 220 - 232: Mere reproduction of the results.

3) Generally, the results of the study were not discussed. It is expected that possible reasons for the findings in the study and how your results compare with similar studies elsewhere should be presented. What are the possible clinical and public health implications of the findings in the study? What conclusions and recommendations can be drawn based on the results of the study?

E) Materials and Methods

1) No information provided on how the isolates were identified.

2) Not clear on what is meant by "double entry conducted on 10% of all data (Line 260).

3) Was the data actually analysed in this study? No results of statistical analysis presented.

4) Line 266: Please check for correct definition of MDR as stated by Magiorakos et al. No date stated!!!!. Magiorakos et al. not listed in the reference list.

5) Sentence on lines 274-275 not clear.   

Reviewer 3 Report

In this article, based on the data of a veterinary laboratory in Hong Kong, the distribution of bacterial species isolated from urinary tract infection in cats and dogs was examined and their antibiotic resistance profiles were evaluated. In particular, the antibiotic resistance distributions of E. coli isolates by years were investigated. Finally, multiple antibiotic resistances were evaluated. AMR is a public health problem that continues to gain importance in every country, therefore it is scientifically very important and the results of the study would help clinicians for choosing the antibiotics for treatment.

There are some points that should be revised.

·         The aim of the study was written as “To contribute to effort against antibiotic resistance, this study aims to investigate and describe the antibiotic resistant pattern in bacteria isolated from urine of feline and canine patients submitted to a veterinary diagnostic laboratory over a  three-year period.” But in discussion the aim of the study is quite changed: “The main objective of this study is to describe and characterize the prevalence of urinary tract bacterial isolates and their antibiotic resistance in canine and feline as companion animals.” But in results sections the authors are focused on E. coli isolates.

Although there is a lot of information throughout the article, the purpose of the study is not clearly stated. Researchers should clearly state the purpose of the study and write the results accordingly, and I suggest evaluating the results related to this purpose in the discussion. (For example; it should be added to the aim of the study to determine the distribution of the most frequently isolated bacterial species. Because there is information and table about it in the results section.

·         In results section, sub-title: 2.1 Feline and canine antimicrobial resistance in commonly identified isolates: In this section, the authors gave the minimum and maximum values, but this causes confusion. Since the exact values are given in the table, it would be more appropriate to write the average isolation rates in this section.

·         Bacteria names should be written in italics both in this section and in the text. The genus name should begin with a capital letter and the species name should continue with a lower-case letter. Also, in some places it is written only as "Enterococcus, Staphylococcus", they should be written as "Enterococccus spp, Staphylococcus spp".

·         Figure-1, Figure-2,  Figure-5, Figure-8, Fıgure9, Figure-10, These are not figures, they should be named as “Table”.

·         In the explanation of Figure-1: Instead of “microbes” it can be more convenient to write as “Bacteria species”.

·         In the Results section, instead of giving exact values, 15 to 20 % (line 97); Values such as about 50% to 60% of these Klebsiella species …( line 104) are written. I can suggest to write more clear/ net values instead of giving variable values like this. Giving too many “approximate” values in a written text negatively affects the quality of the text. A single value can be given by taking the average in the text and the details can be given clearly in the table.

·         It is very difficult to read Figure 3 and 4. And the information about those figures are complicated and difficult to follow. It would be better to review these paragraphs.

Figure-9 and 10 could be removed since these results have not been discussed. Or there should be some points in the discussion section about those MDR profiles. For example; the authors has mentioned many patterns of MDR. Clinically important profiles can be mentioned in the discussion section.

·         Are the MDR results include E. coli isolates?

·         Why only E. coli isolates were focused in “Temporal trend of isolates’ antibiotics resistance in feline and canine species

·         References: Lines 53-68: The authors provide information about antibiotic use guidelines and about empirical treatment in urinary tract infections. However, the cited reference relates to upper respiratory tract infections. I suggest the authors check the references.

There should be a reference for “first-line and non-first-line antibiotics” in the manuscript.

If the article is revised according to these suggestions, I think that it will be an easier to follow and more understandable article. In this way, clinicians will be able to see important resistance profiles more easily.
